# At the Crossroads of the cGAS-cGAMP-STING Pathway and the DNA Damage Response: Implications for Cancer Progression and Treatment

**DOI:** 10.3390/ph16121675

**Published:** 2023-12-01

**Authors:** Tatyana V. Korneenko, Nikolay B. Pestov, Ivan A. Nevzorov, Alexandra A. Daks, Kirill N. Trachuk, Olga N. Solopova, Nickolai A. Barlev

**Affiliations:** 1Group of Cross-Linking Enzymes, Shemyakin-Ovchinnikov Institute of Bioorganic Chemistry, Moscow 117997, Russia; 2Institute of Biomedical Chemistry, Moscow 119121, Russia; 3Chumakov Federal Scientific Center for Research and Development of Immune-and-Biological Products, Moscow 108819, Russia; 4Institute of Cytology, Tikhoretsky ave 4, St-Petersburg 194064, Russia; 5Research Institute of Experimental Diagnostics and Tumor Therapy, Blokhin National Medical Research Center of Oncology, Moscow 115478, Russia; 6Institute of Translational Medicine and Biotechnology, Sechenov First Moscow State Medical University, Moscow 119991, Russia

**Keywords:** STING, cGAS, cGAMP, interferon signaling, cancer, metastasis, BRCA, proton channel

## Abstract

The evolutionary conserved DNA-sensing cGAS-STING innate immunity pathway represents one of the most important cytosolic DNA-sensing systems that is activated in response to viral invasion and/or damage to the integrity of the nuclear envelope. The key outcome of this pathway is the production of interferon, which subsequently stimulates the transcription of hundreds of genes. In oncology, the situation is complex because this pathway may serve either anti- or pro-oncogenic roles, depending on context. The prevailing understanding is that when the innate immune response is activated by sensing cytosolic DNA, such as DNA released from ruptured micronuclei, it results in the production of interferon, which attracts cytotoxic cells to destroy tumors. However, in tumor cells that have adjusted to significant chromosomal instability, particularly in relapsed, treatment-resistant cancers, the cGAS–STING pathway often supports cancer progression, fostering the epithelial-to-mesenchymal transition (EMT). Here, we review this intricate pathway in terms of its association with cancer progression, giving special attention to pancreatic ductal adenocarcinoma and gliomas. As the development of new cGAS–STING-modulating small molecules and immunotherapies such as oncolytic viruses involves serious challenges, we highlight several recent fundamental discoveries, such as the proton-channeling function of STING. These discoveries may serve as guiding lights for potential pharmacological advancements.

## 1. Introduction

The DNA-sensing cGAS-cGAMP-STING innate immunity pathway was originally discovered in 2008–2013 [1,2,3,4,5,6,7,8] and has attracted widespread attention in respect to its role in tumorigenesis [9,10,11,12,13]. The number of articles on this topic has grown explosively (232 Pubmed entries in 2013–2017 and 1334 in 2018–2022), including excellent recently published reviews [14,15,16,17,18,19,20,21]. The cGAS-STING pathway senses cytosolic DNA and becomes activated in response to viral invasion. Mechanistically, the cGAS-STING pathway (a name hereinafter referring to the canonical cGAS-cGAMP-STING-TBK1-IRF3,7 pathway) transduces a signal from pathogen associated molecular pattern (PAMP) molecules, e.g., fragments of DNA, to specific effector proteins such as the transcription factors IRFs (interferon responsive factors) and NF-kB (nuclear factor κB). These transcription factors trigger the expression of pro-inflammatory genes such as interferon (IFN), thereby eliciting an immune response using both autocrine and paracrine mechanisms. The classical action of IFN involves the upregulation of many other genes, such as 2′-5′-oligoadenylate synthetase (OAS), which in turn activates ribonuclease (RNase L) to halt viral replication.

Theoretically, recognition of either foreign non-self (e.g., infections) signals or aberrations within the endogenous self (e.g., malignancies) relies on the organism’s friend-or-foe recognition system, which must be meticulously calibrated to fight infections and tumors without inflicting harm to healthy tissues. This is why pattern recognition receptors (PRRs) are fundamental to the innate immune system.

Various forms of nucleic acids can be perceived as foreign or malignant. Whether it is naked DNA, modified or unmodified sequences, DNA coated with histones, DNA-RNA hybrids, RNA, small fragments, or cyclic dinucleotides, each can be seen as foreign elements to different extents. The cGAS-STING pathway is extremely important for these functions, but in fact, the cellular sensors of DNA are much more diverse.

The human genome contains over 14 identified DNA sensors, among which three—TLR9, AIM2, and cGAS—are typically considered the primary players [22]. Several others are specifically important for the subject of our review:(1)TLR9 (Toll-like receptor 9) is found almost exclusively in the endosomal membrane of B cells and plasmacytoid DCs (dendritic cells). It detects external DNA and is particularly sensitive to unmethylated or hypomethylated CpG islands [23];(2)IFI16 and AIM2 belong to the PYHIN family [24]. IFI16 detects both DNA and RNA. It predominantly resides in the nucleus and cytosol and primarily detects DNA strands around 70 base pairs in length. Upon activation, IFI16 stimulates the inflammasome. It is expressed in many cell types and is particularly abundant in leukocytes. AIM2, located in the cytosol, is an active defender against viruses and bacteria. When activated, it facilitates the formation of the inflammasome, which can trigger the activation of caspases. This activation leads to the release of cytokines IL-1β and IL-18 and induces pyroptosis, especially in response to DNA damage caused by acute ionizing radiation. AIM2 exhibits high specificity for T and B cells, as well as spermatids;(3)DDX41 senses dsDNA, ssDNA, cyclic di-GMP, and cyclic di-AMP. This detection results in the transcription of genes crucial for the innate immune response. Notably, in chickens, DDX41 has demonstrated a strong interaction with STING [25,26];(4)ZBP1 is highly enriched in immune cells (plasma cells, NK cells) and is a sensor for dsDNA and dsRNA helices in the left-handed Z conformation, namely, Z-DNA and Z-RNA [27] ZBP1 plays a key role in mediating various forms of cell death. Its activation leads to RIPK3 stimulation and subsequent MLKL phosphorylation, causing disruption of the nuclear envelope and release of cellular DNA into the cytosol. ZBP1 stabilizes Z-form mtDNA and forms a cytosolic complex with cGAS;(5)Among the cytosolic DNA sensors, however, cGAS appears to be one of the most widespread and important at most stages of infection (Figure 1).

Unsurprisingly, the cGAS-STING pathway was shown to intertwine with the DNA damage response system because both systems deal with fragmented DNA. However, many fundamental mysteries of this interplay are just beginning to be revealed. For example, are there distinguishing characteristics between viral DNA and cellular DNA that are essential for initiating the cascade? Why is cGAS-STING signaling not activated by cGAS in the nucleus or by the cell’s own DNA in the cytoplasm—of which there is an abundance during cellular aging? 

In this review (key references collected from 2008 to September 2023), we attempt to discuss the mechanisms of the interactions between cGAS-STING and cancer progression with the purpose of envisaging approaches to the development of novel cancer treatment modalities that exploit this unique pathway.

## 2. Overview of the Classical cGAS-cGAMP-STING-TBK1-IRF3 Pathway: Structure of Key Components and Their Molecular Evolution

The cGAS-STING axis is a typical cellular signaling pathway in terms of its complexity and interconnections with other regulatory networks, yet it has certain remarkable features, such as the unique enzymatic activity of cGAS.

### 2.1. cGAS (Cyclic GMP-AMP Synthase)

This enzyme is involved in several physiological functions, predominantly infection defense and associated inflammatory responses [28]. cGAS belongs to the large cGAS/DncV-like (dinucleotide cyclase in *Vibrio*) nucleotidyltransferase family and has deep phylogenetic roots [29,30,31]. Enzymes with similar activity have been identified in both prokaryotes and eukaryotes, although bacterial counterparts produce a wide variety of cyclic di- and oligonucleotides.

cGAS is able to interact with sufficient affinity with double-stranded DNA longer than 20 bp [5,32]. This length reflects the minimum DNA-binding requirement for the cGAS monomer. However, for optimal activation, cGAS typically requires DNA strands longer than 45 bp to establish stable enzymatic complexes—dimers of DNA-cGAS (Figure 1). Binding of cGAS to DNA is also strongly enhanced when the DNA is bent. Such bending is positively influenced by HMGB1 (High Mobility Group B1) protein [33]. cGAS works in a sequence-independent fashion, but there are exceptions, like its interaction with some specific viral ssDNA (single-stranded DNA) structures in HIV-1 (Human Immunodeficiency Virus type 1), as demonstrated in HEK293T cells [34].

cGAS can be activated not only by dsDNA (double-stranded DNA), but also by RNA:DNA hybrids, a unique trait among nucleic acid sensors involved in IFN induction, as shown in murine cells [35]. These hybrids, referred to as R-loops, also stimulate cGAS, and subsequently, IRF3, culminating in apoptosis, especially in BRCA1 (BReast CAncer susceptibility gene 1)-mutated cancer cells in vitro [36,37].

By contrast, cGAS complexes with dsRNA are formed without proper conformational change and result in no activation and thus in no efficient induction of IFN. cGAS in the cytoplasm may form transient low-affinity complexes with RNA. Moreover, some RNAs are strong inhibitors of cGAS; for example, in hematopoietic stem cells, circular RNA prevents cGAS activation by nuclear self dsDNA, maintaining stem-cell dormancy (shown in vitro) [38]. As some RNA viruses are also susceptible to cGAS-STING, it is tempting to speculate that RIG-I-like (Retinoic acid-Inducible Gene I) receptors are not the sole players in this field. One should anticipate further discoveries of regulatory roles for non-coding RNAs [39]. Recent studies indicate, however, that cytoplasmic RNAs do not activate cGAS per se, but rather enhance its phase separation into liquid-liquid condensates [40,41]. Most likely, activation of cGAS by RNA is a result of reverse transcription, as it is in the case of retroviruses like HIV-1 [42].

Not only unscheduled DNA release, but also other cellular stresses may be operational in terms of activation of cGAS. Perturbations of mRNA translations such as frameshifts are quite common during viral infections; indeed, a recent discovery describes activation of cGAS by pure ribosomes, and this activation may be reminiscent of ribosome collision in vivo. However, in this case, activation of cGAS do not proceed to IFN secretion [43].

Structurally, human cGAS contains three distinct DNA binding sites within its C-terminal domain [28,34,44,45]. Its N-terminal domain, intrinsically disordered in the absence of DNA, appears to be non-essential for recognition of pathogenic or mitochondrial DNA [46,47,48,49,50], but it can give additional stability to the cGAS-DNA complex [34,51]. This domain also mediates interactions with membranes and nuclear DNA and is important for cGAS enrichment in centromeres. Additionally, cGAS has an N-terminal core domain and a Mab21 domain, the latter of which contains a zinc-ribbon domain that is vital for DNA binding. 

In the absence of DNA, cytoplasmic cGAS might be anchored to proteins such as Ras-GTPase-activating protein-binding protein 1 (G3BP1). This possibility hints at a possible displacement by DNA in its pre-condensed state. Notably, G3BP1 augments the DNA binding capability of cGAS [52].

### 2.2. cGAMP (Cyclic GMP-AMP)

When cGAS is activated by DNA, it synthesizes the secondary messenger 2′,3′-cyclic CMP-AMP (2′,3′-cGAMP) [2,53,54], a member of the cyclic dinucleotide (CDN) family. CDNs are important and diverse signaling molecules in prokaryotes that are involved mostly in anti-phage defense [55]. The mammalian cGAMP is 2′,3′-cGAMP, a unique molecule with two distinct phosphodiester linkages: 1) 2′-OH of GMP to 5′-phosphate of AMP; and 2) 3′-OH of AMP to 5′-phosphate of GMP.

### 2.3. STING (Stimulator of Interferon Genes)

STING apparently originated early in bacteria, but the degree of sequence conservation is low (only 8 hyperconserved residues). Moreover, bacterial STINGs bind c-di-GMP with high affinity, preferring 3′–5′-linked cyclic dinucleotides, in contrast to human STING. Bacterial STINGs are not activated by mammalian 2′,3′-cGAMP [56]. In contrast, mammalian STING can also act as an independent pattern recognition receptor for bacterial cyclic dinucleotides (this has been demonstrated using *Listeria monocytogenes* infection models in C57BL/6 mice and HEK293T cells) [57,58]. The observable effects of other CDNs from diverse organisms on mammalian STING may not be limited to 2′,3-cGAMP, as has been shown in vitro in mammalian cells [59]. Therefore, the mechanism of STING activation by CDNs is conserved across various organisms. 

Structurally, the product of the STING1 gene (a.k.a. ERIS, FLJ38577, MITA, MPYS, NET23, STING, TMEM173) is a multi-domain membrane protein featuring four transmembrane α-helices with only small loops exposed into the ER (Endoplasmic Reticulum) lumen, a cGAMP binding domain, and another cytoplasmic domain required for recruitment of downstream signalling molecules. Mammalian STING is activated by binding either c[G(2′,5′)pA(3′,5′)p] or its isomer c[G(2′,5′)pA(2′,5′)p] [60]. However, only 2′,3′-cGAMP is considered to be fully functional in terms of STING stimulation [61]. Binding of cGAMP to STING leads to its activation by increasing the level of STING oligomerization: conversion from an inactive dimer to an active tetramer and higher-order oligomers [62]. Oligomers dissociate easily, but certain small molecules can adjust the stability of the oligomers [63]. In turn, this process of oligomer formation can be deregulated in various ways; for example, a single mutation can render STING perpetually active [64]. 

More specifically, in its resting state in the ER, STING exists as a dimer, complexed with proteins like TRAPβ, SEC61β, and STIM1. Upon binding to 2′,3′-cGAMP, the wing-like structure of STING undergoes a remarkable movement, accompanied by the rotation of the ligand-binding domain, and this facilitates its translocation from ER to Golgi, where it binds to sulfated glycosaminoglycans [62,65,66,67] and may open a proton pore [68]. Activation of STING by oligomerization is also promoted by many other proteins, including the mitochondrial nucleoid proteins HU, mitochondrial transcription factor A (TFAM), and HMGB1 (the highly mobile group box protein 1) [33] (various proteins mentioned in this review are listed in Appendix A). The extensive oligomerization of the cGAS-DNA complex [51] also gives rise to observable liquid-liquid condensation (phase separation), particularly in conjunction with STING oligomers during their transition from the endoplasmic reticulum (ER). This effect can be seen in cultured cells like HEK293T, COS-7, HeLa, and THP-1 when they are exposed to viral DNA [69]. The formation of small intracellular condensates containing nucleic acids and proteins is now being extensively studied [70]. 

The transport of STING from the ER relies heavily on interactions with coat protein complex II (COPII), adenosine diphosphate ribosylation factors (ARF) GTPases, and palmitoylation at the transmembrane cysteine residue 91. These interactions are a critical step for its assembly into multimeric complexes at the Golgi (as demonstrated in various cell lines and in agonists1-deficient mice) [71]. For effective downstream signaling, both the formation of STING oligomers and their transport are essential.

### 2.4. TBK1 (TANK-Binding Kinase 1)

The next indispensable player in the pathway is TBK1 [59]: in proximity to the Golgi, TBK1 phosphorylates the STING C-terminus, enabling its interaction with IRF3 and IRF7. Consequently, IRF3 or IRF7 is trapped, leading to its inevitable phosphorylation by TBK1. This interaction highlights the central role of TBK1 in the pathway. Additionally, when this complex is formed, TBK1 undergoes auto-phosphorylation. TBK1 inhibits this oligomerization process in cells treated with 3′-cGAMP as a mode of regulation [72,73] TBK1 shows enhanced activity when STING is polyubiquitinated, a process mediated by E3 ubiquitin ligase—autocrine motility factor receptor (AMFR) combined with insulin-induced gene 1 (INSIG1) (for example, this mediation is observed in cultured cell lines such as L929, THP, and HEK293 [74]). 

Together with IκB kinase (IKKβ), TBK1 plays a crucial role in STING signaling, activating both IRF3 (interferon regulatory factor 3) and NF-κB. Therefore, STING-dependent activation of TBK1 leads to the concurrent activation of both IRF3 and NF-κB pathways, in a process akin to a fork in the signaling route [75]. 

### 2.5. IRF3, NF-κB and Downstream Signaling 

The activation of cGAS, STING, and TBK1 ultimately leads to the activation of transcription factors, including IRF3, IRF7, and NF-κB. These factors regulate the production of pro-inflammatory cytokines such as IFN1α, IFN1β, TNFα, IL-6, IL-12, and IL-1β (exemplified in cells like HEK 293T and L929 [76,77]). Mechanistically, the phosphorylated IRF3 dimerizes, dissociates from the complex, and relocates to the nucleus, activating IFN type I, particularly IFN-β. The secreted IFN modulates the activity of other cells, activating IFN-α/β receptor 1 and 2 (IFNAR1/2), which subsequently triggers the signal transducer and activator of transcription (STAT) pathway as an important part of antiviral defense. NF-κB activation via STING involves various proteins, including NF-κB essential modulator (NEMO), IKKβ, TBK1, and tripartite motif-containing protein 32 and 56 (TRIM32/56) [78,79]. 

In a straightforward model of cGAS-STING activation through dsDNA transformation, marked upregulation of pTBK1, pIRF3, and pSTING can be observed. In contrast, there might be a depletion of total STING levels. Additionally, among the IFN-responsive genes, a robust upregulation of IFNβ, OAS1, C-X-C motif chemokine ligand 10 (CXCL10), and interferon-stimulated gene 15 (ISG15) can be detected [80]. Transgene expression, following transfection with naked DNA, may be augmented by inhibition of cGAS-STING [81]. Also, the efficiency of DNA transfection by electroporation can be enhanced by inhibition of cGAS activity, as the latter boosts cell viability (shown in mice and in vitro in Nalm6 and THP1 cells) [82].

The alternative or ‘non-canonical’ STING pathway is coordinated by IFI16 (Interferon γ-Inducible Protein 16), in collaboration with DNA damage response elements such as ATM (Ataxia Telangiectasia Mutated) and PARP-1 (Poly(ADP-Ribose) Polymerase 1) (Figure 2). In this case, STING signaling also engages the tumor suppressor p53 and the E3 ubiquitin ligase tumor necrosis factor receptor (TNFR)-associated factor 6 (TRAF6), which polyubiquitinates STING at K63. This sequence of events leads to the activation of NF-κB (shown in vitro in HaCaT, MRC-5, and HEK293T cells) [83,84]. Additionally, in triple-negative breast cancer (TNBC), IFI16 has been observed to accumulate at sites of double-stranded DNA breaks, enhancing the release of DNA fragments into the cytoplasm and triggering STING-mediated type-I IFN production [85]. 

## 3. cGAS-STING in Major DNA Damage Response Pathways 

It is pertinent to note that the cGAS-STING pathway is intricately intertwined with numerous signaling pathways, including those involved in DNA damage, rendering it susceptible to varied modes of regulation. For instance, in the event of DNA damage, the activation of the ATM/Chk2 (checkpoint kinase 2) pathway induces a replication checkpoint within the cell. Indeed, pharmacological inhibition of ATM imposes replication stress, leading to an increased abundance of DNA fragments in the cytosol and hence to STING activation [86]. Moreover, STING is the major adaptor protein of the DNA sensors, as it exists at the convergence of the activities of cGAS, IFI16, ZBP1 (Z-DNA-binding protein 1)/DAI (DNA-dependent activator of interferon-regulatory factors), DDX41 (DEAD (Asp-Glu-Ala-Asp)-box polypeptide 41), MRE11/Rad50, and Ku 70 [1,57,87,88]. Thus, STING engages in direct cross-talk with other pathways, for example, in myeloid cells MYD88 (Myeloid Differentiation Primary Response 88). STING can also form a direct liaison [89], making the DNA-sensing network quite complex, as MYD88 also interacts with β-catenin and activates the aforementioned IRF3, IRF7 and NF-κB. 

### 3.1. Efficient Oncosuppressive Function of cGAS-STING in DNA Damage Response 

Functional cytoplasmic DNA sensing by the cGAS-STING pathway significantly contributes to tumor suppression on the organismal level [90]. While the absence of cGAS or STING does not directly induce carcinogenesis, it is clear that the action of innate immunity precedes the priming of tumor antigen-specific CD8+ T cells. The cGAS-STING pathway serves among other mechanisms in numerous anti-cancer checkpoints. For example, cGAS knockout makes mice susceptible to colitis-associated colorectal cancer due to disruption of intestinal barrier integrity, decreased stem cell number, inflammation with abundant myeloid-derived suppressor cells (MDSC), and STAT3 activation. Interestingly, normal intestinal epithelium appears to contain cGAS in a clear excess to STING, as shown in vivo in an experiment including cGAS^−/−^ mice as controls [91]. STING knockout mice showed no change in the initial onset of hepatocellular carcinoma but significantly more pronounced tumor growth in the later stages [92]. Similarly, in mouse models of colorectal carcinoma, this mechanism allows cancer cells to evade the immune response [93]. 

A meta-analysis encompassing various human cancers suggests that the presence of STING in tumors is most often associated with a favorable prognosis, correlating with increased disease-free survival/recurrence-free survival (DFS/RFS) but not with overall survival (OS) [94]. 

In discussing the regulation of the cGAS-STING axis, we should stress again that STING knockout results in heightened sensitivity to viral or bacterial pathogens and that this function is evolutionary conserved [95]. In oncology. However, this situation is much more complex and intricate. 

### 3.2. Senescence as the Route out of Cancer and the Role of the STING Pathway

Senescence and short-term inflammation are powerful barriers to tumorigenesis, whereas persistent inflammation is associated with tissue damage, and in established cancers, is linked to tumor growth and metastatic dissemination. This linkage is confirmed by the observation that cGAS-triggered senescence occurs upon irradiation and oncogene activation in mice [96], supporting the hypothesis that the senescence-associated secretory phenotype (SASP) controls tumorigenesis. cGAS is localized in the cytoplasm of non-dividing cells but enters the nucleus and associates with chromatin DNA during mitosis in proliferating cells [97]. 

Evading senescence can be a pathway to cancer, and the replicative crisis serves as a protective mechanism. It is thus not surprising that cGAS also senses telomeric damage. Specifically, extrachromosomal telomere repeats (ECTR) DNA is common to cancer cells that utilize the alternative lengthening of telomeres (ALT) pathway. ECTRs in typical human fibroblasts activate the canonical cGAS-STING pathway, causing proliferation defects. Conversely, ALT cancer cells often have defective cytosolic DNA sensing. STING expression is reduced in ALT cancer cell lines and transformed ALT cells [98]. Mice with cGAS or STING knock-out (KO) show no SASP in response to IR or RasV12 (in normal mouse RasV12 cells enter SASP and are killed, whereas and in cGAS or STING KO these cells are retained) [99]. Recently, the cGAS–STING pathway was found to play a significant role in driving inflammation related to aging through microglial intrinsic involvement in age-associated neurodegeneration. In mice, the activation of cGAS alone initiates a fundamental microglial gene expression program, which is common across various neurodegenerative diseases [100]. 

### 3.3. DDR and Its Links to Carcinogenesis and Tumor Formation

Certain viruses, like parvoviruses, exploit DNA damage response (DDR) pathways. Consequently, it is plausible to suggest that antiviral defenses could counter DDR mechanisms. Supporting this hypothesis, cGAS has been observed to impede DDR, particularly targeting the homologous recombination (HR) pathway, while not affecting the non-homologous end joining (NHEJ) pathway [101]. These disturbances can be categorized into several phases: presence of abnormal DNA in the nucleus, such as extrachromosomal DNA and extrachromosomal telomeric repeat DNA; loss of nuclear-envelope integrity; formation of micronuclei; rupture of micronuclei; creation of DNA-containing vesicles; cytoplasmic chromatin fragments; presence of senescence-associated DNA in the cytosol; formation of abundant abasic DNA. Micronuclei are characteristic small nucleus-like bodies composed of fragments of chromosomes surrounded by fragile nuclear envelopes (Figure 3). They can be generated during mitosis from chromatid fragments formed by DNA double-strand breaks (DSBs) or from lagging chromosomes formed by mis-segregation [102].

The formation of a significant number of micronuclei is associated with escape from mitotic block, defects in chromosome segregation, and accumulation of damaged DNA fragments. In normal cells, any anomalies (exemplified by deletion of actin-related protein 2/3 complex (ARP2/3) [103]) in vitro in various cell lines) lead to activation of p53 that halts the cell cycle in G1 through cyclin-dependent kinase inhibitor 1A (CDKN1a) and p21, and any significant DNA accumulated in cytosol induces the cGAS-STING- and IRF3-associated IFN response. The so-called CAD (caspase-activated DNase) participates in micronucleus formation, as its inhibition reduces micronucleus formation and metastasis, while CAD activation, by contrast, results in a more aggressive phenotype in a STING-dependent manner [104]. The survival of cancer cells that have accumulated micronuclei is an intriguing subject and is currently the focus of active research. cGAS fully activates upon rupture of micronuclei [105]; however, specific conditions preceding this rupture can affect the activation process. For instance, if DNA transcription within the micronuclei is active, the activation of cGAS is less efficient [106]. Mitotic arrest can be induced by nocodazole, resulting in chromosomal instability (CIN) with abundant micronuclei and aberrant chromosome segregation, whereas decreased p21 (caused by depletion of STING, TBK1, IRF3 or by overexpression of cGAS) inhibits these phenomena (in vitro in hTERT-RPE1 and cGAS-deficient HeLa cells) [107]. Similarly, activation of cGAS and IRF3 phosphorylation enhance sensitivity to taxol in xenografts [108].

Mutations in p53 can antagonize STING by obstructing the binding between STING and TBK1. Specifically, mutant p53, unlike wild-type p53, interacts with TBK1 and inhibits the formation of the complexes involving TBK1, STING, and IRF3 (in vitro in various cancer lines such as MDA-MB-231, BT549, H1299, MiaPaCa-2, and A549, and in vivo in BALB/c and NOD/SCID xenografted mice) [109,110].

Interestingly, non-canonical NF-κB is not only activated in some cancer cells, but boosts expression of genes such as RAD51 that are responsible for HR [111].

### 3.4. STING and BRCA1,2

BRCA1 is essential for DNA repair (including repair of transcription-associated DNA damage). When it is not degraded by exonucleases such as MRE11 (Meiotic Recombination 11 Homolog 1), single-stranded DNA from halted replication forks translocates into the cytosol and activates the cGAS-STING pathway, signaling a response to replication stress [37]. Interestingly, cGAS binds directly to the replication fork, decelerating replication but reinforcing the DNA’s stability. Without cGAS, there is increased sensitivity to both radiation therapy and chemotherapy [112]. Fascinatingly, BRCA1-mutated cancer cells display accumulation of cytoplasmic RNA-DNA hybrids, which are products of R-loop processing. Such accumulation, sensed by cGAS and TLR3, is apparent when the nuclear R-loop metabolism becomes unregulated. Notably, this event is prominent in cells with mutations in SETX and BRCA1 [36].

Equally intriguing is the role of the hyaluronan-mediated motility receptor (HMMR, also known as RHAMM). Its interaction with BRCA1 has ramifications not just for cell division, but also for the polarization of certain epithelial cells, influencing cell division in the process. Mouse models have shown how HMMR magnifies the pro-oncogenic effects of mutated BRCA1. This interaction leads to the formation of micronuclei and subsequent induction of cGAS-STING, followed by activation of both canonical and non-canonical NF-κB signaling pathways [113].

It is important to distinguish between intrinsic and extrinsic tumor processes in cancer cells. While BRCA1 functions alongside PALB2 (Partner and Localizer of BRCA2), disruption of their interaction in a hepatocellular carcinoma mouse model led to heightened T-cell infiltration and enhanced programmed cell death protein 1 (PD-1) antibody response, in sync with both intrinsic and extrinsic cGAS-STING activation [114].

BRCA2 inactivity, similarly to inactivity of BRCA1 or FANCD2 (Fanconi Anemia Group D2 Protein), escalates TNFα production, leading to augmented autocrine TNFα signaling. The absence of BRCA2 is also associated with the formation of micronuclei and activation of the apoptosis signal-regulating kinase 1 (ASK1) and c-Jun N-terminal kinase (JNK) pathways. BRCA2-deficient triple-negative breast cancers (TNBCs) also may show some intensification of cGAS-STING signaling (in cell lines like KBM-7, BT-549, HCC38, and MDA-MB-231) [115]. 

It is worth emphasizing that essential genome integrity signaling operates independently of the canonical IFN pathways. This independence is highlighted by the fact that cGAMP reduces NAD^+^ levels, subsequently inhibiting PARylation and suppressing HR [116]. On the other hand, activation of the cGAS-STING-TBK1 axis prompts DDR induction through the autophosphorylation of ATM kinase. This action triggers the CHK2-p53-p21 signaling pathway, resulting in cell cycle arrest in murine models [116]. 

### 3.5. STING and Oncogenic Infections

Chronic *H. pylori* infection is strongly associated with increased risk of noncardia gastric cancer. It is thus rather interesting that *H. pylori* infection can increase STING expression and activate its signaling in mice [117]. It is not yet clear how normal microflora affect the cGAS-STING pathway in colorectal cancer progression, but interesting data link STING to colitis and colitis-associated cancer (CAC)—STING knock-out in mice reduces pyroptotic cell death, making the animals more prone to CAC [118] (compare this to the different results of cGAS knockout [91]). 

Oncogenic viruses obviously have to contend with host innate immunity and nucleic-acid sensors to exert their pathogenic effects. Around 12% of all human tumors are caused by such viruses, and the cGAS-STING pathway should play a prominent role as an anti-cancer guardian against at least some of these viruses. Conversely, these viruses should suppress this pathway. Indeed, the E1A and E7 oncoproteins of human papillomavirus (HPV), the main etiologic factor in cervical cancer, were shown to directly interact with STING via the LXCXE motif and inhibit cGAS-STING signaling [119]. Additionally, Kaposi sarcoma-associated herpesvirus (KSHV) produces the vIRF1 protein, which interacts with TBK1 and inhibits STING phosphorylation [120]. Targeting viral proteins involved in the suppression of the cGAS-STING pathway remains an underexplored approach for the treatment of tumors induced by viral infections. 

However, the IFI16-STING axis may play a pro-cancer role in the context of cancers induced by oncogenic viruses. Notably, there is significantly higher expression of PD-L1 (Programmed cell Death 1 Ligand 1), and IFI16 in HPV positive cervical cancer cells. The sequence of events includes IFI16-activated STING-TBK1-mediated activation of the NF-kB pathway, which, in turn, enhances PD-L1 expression [121].

## 4. Cross-Talk of cGAS-STING with Major Cell Death Pathways

The roles of cGAS and STING in cell death pathways are well-documented and reviewed [122]. Generally, STING activation per se does not induce cell death, either necrotic or PCD (programmed cell death); only some cells are susceptible, like human myeloid cells in vitro [123]. In T cells, naïve B cells, and malignant B cells, STING or its agonists can trigger mitochondrial apoptosis; this phenomenon was especially well documented in murine models of chronic lymphocytic leukemia and multiple myeloma [124]. The role of cGAS-STING in apoptosis has been extensively reviewed recently [125]. In our view, other cGAS-STING cell-death pathways are more interesting in the oncological context (Figure 1 and Figure 2).

### 4.1. Necroptosis and cGAS-STING 

Necroptosis, or programmed necrosis, dependent on the action of RIPK3 (Receptor-Interacting Protein Kinase 3) or MLKL (Mixed Lineage Kinase Domain-Like Protein), requires a basic level of IFN signaling. cGAS-STING plays an important role in necroptotic cell death in cancers. Activation of necroptosis by RIPK3 or MLKL in macrophages is dependent on DNA sensing and is elevated in autoimmune conditions like systemic lupus erythematosus (SLE) [126]. Necroptotic signaling stimulates STING, whereas suppression of RIPK3 inhibits STING by redirecting STING to autophagosomes [127]. In mouse models, STING agonists induce necroptosis in primary macrophages in vitro in a STING and RIPK3-dependent pathway linked to production of and IFN and TNF [128]. 

### 4.2. Pyroptosis and cGAS-STING

Pyroptosis is the cell death pathway that employs the NLRP3 (NLR family pyrin domain-containing 3) inflammasome and the formation of plasma membrane pores by gasdermin, as well as lysosomal cell death. These pathways are promoted by STING after its translocation to the lysosome through an increase in lysosomal membrane permeabilization. Especially in human myeloid cells, the cGAS-STING-NLRP3 pathway works as a default death program [123]. Also, spleen tyrosine kinase (SYK) was found to be the key mediator of pyroptosis and also a direct interactor of STING [118]. A significant recent finding is that STING activation causes a marked increase in pH within the Golgi, attributed to STING’s function as a proton channel. This channel activity is essential for subsequent NLRP3 inflammasome activation, leading to pyroptosis and IL-1β secretion. This process is distinct from the lipidation and autophagy processes, as shown in mice [68,129].

### 4.3. Ferroptosis and cGAS-STING

Activation of STING stimulates ferroptosis, the cell death pathway promoted by iron-dependent oxidation of lipids and other intracellular components in the presence of low levels of gluthatione peroxidase, and this effect of STING is especially noticeable in human pancreatic cancer cells [130]. Several mechanisms might be at play, including those involving reactive oxygen species (ROS) or lipid peroxidation. For example, the ferroptosis inducer erastin enhances accumulation of STING in mitochondria, increasing mitochondrial fusion concomitantly with increases in ROS and lipid peroxidation, and in xenografted mice, STING KO renders these cells less prone to ferroptosis induction (in NOD/SCID mice with STING1-KD or MFN1/2-KD PANC1-derived xenograft tumor) [130]. Broadly, the mechanism of such STING action may involve ROS that act through 4-hydoxy-nonenal, promoting STING carbonylation at C88 and thus preventing palmitoylation and translocation from the ER. Thus, ROS action is predominantly STING-inhibiting and may decrease responses against both pathogens and tumors [131]. Additionally, STING may facilitate ferroptosis in the kidney. One mechanism for this phenomenon is STING-dependent ferritinophagy, which occurs through interaction with nuclear receptor coactivator 4 (NCOA4). In this scenario, ferroptosis, characterized by increased ROS and lipid peroxidation, may be induced by STING expression or inhibited by STING antagonists in mouse models [132]. 

### 4.4. Autophagy and cGAS-STING

STING contributes to autophagy, facilitating the removal of cytoplasmic DNA. This process results in a reduction of cGAS levels, while beclin 1 acts to inhibit DNA-cGAS. This intricate regulatory loop operates independently of IFN and TBK1 activation and is mediated by lipidated LC3 on the ER-Golgi intermediate compartment (ERGIC) and the Golgi, in a mechanism reliant on the COPII complex and ARF GTPases [129]. The ERGIC that contains STING functions as a membrane source for LC3 lipidation, a critical phase in autophagosome formation. cGAMP triggers LC3 lipidation via a pathway dependent on WIPI2 (WD repeat domain phosphoinositide-interacting protein 2) and ATG5 (autophagy-related protein 5) but independent of Unc-51-like kinase (ULK) and VPS34-beclin kinase complexes.

## 5. Roles of the cGAS-STING Pathway in Response to Radio- and Chemotherapies

### 5.1. Radiotherapy Elicits DNA Damage and Activates cGAS-STING

The cGAS-STING pathway is important in the response to radiotherapy that relies on activation of DCs. In this schematic, a critical phase occurs when ionizing radiation (IR) provokes a surge in cGAMP secretion from tumor cells (Figure 4). This escalation subsequently induces the release of type I interferon in adjacent cells. This response, which is crucial to the adaptive immune response to radiation, relies predominantly on the cGAS-STING pathway, rather than on HMGB-1 release or MyD88 signaling in mice [133]. cGAMP, acting through STING, initially induces TNFα-dependent hemorrhagic necrosis in a manner that is independent of T-cells. Subsequently, it aids in the CD8 T-cell-dependent containment of any residual disease, counteracting the tumor-promoting effects of M2 macrophages in pancreatic cancer in mice [134].

Interestingly, the non-canonical NF-κB pathway in dendritic cells (DCs) has been shown to interfere with the anti-tumor activity of cGAS-STING during radiotherapy [135]. In some scenarios, STING activation during radiotherapy might be immunosuppressive. Crucially, precise targeting of the CCR2 pathway can potentially negate this undesirable effect by inhibiting immunosuppressive MDSC (myeloid-derived suppressor cell) infiltration [136]. 

This duality could be attributed to the fact that the cGAS-STING pathway is not the only contributor to radiotherapy-induced antitumor immunity. For instance, ZBP1-dependent activation of MLKL amplifies type I interferon responses following irradiation: the ZBP1-MLKL necroptotic cascade leads to accumulation of cytoplasmic DNA, which autonomously activates cGAS-STING signaling, and this positive feedback loop drives persistent inflammation [137].

### 5.2. Chemotherapy and cGAS-STING

Pharmaceutical agents such as cisplatin and etoposide can induce leakage of nuclear DNA into the cytoplasm, inherently triggering STING-dependent cytokine production. Subsequently, these inflammatory cytokines undergo external amplification through a STING-dependent process, which is facilitated by infiltrating phagocytes that target the dying cells [138]. Furthermore, in tumor-bearing mice, cGAMP increases the antitumor activity of 5-FU (5-flurouracil) and also reduces its toxicity in mice with colon adenocarcinoma [139]. 

The cGAS-STING pathway plays a significant role in the response to other DNA-damaging agents. For instance, cGAS-STING signaling amplifies the immune response induced by cisplatin in patients with bladder cancer. In a mouse xenograft tumor model, induction of cGAS-STING signaling by cisplatin enhanced CD8+ T-cell and dendritic cell infiltration and thus reduced bladder cancer cell growth [140]. In BRCA1-deficient breast cancers, cisplatin also acts concomitantly with STING activation [115,141].

### 5.3. PARPi and cGAS-STING

cGAS co-localizes with PARP1 and, by impeding DDR, prevents specific viruses from exploiting this mechanism (as noted in the Section 3.3). Notably, olaparib, a characteristic PARP inhibitor (PARPi), obstructs the interaction between PARP1 and cGAS [101]. PARPi induces accumulation of cytosolic DNA fragments due to unresolved DNA damage, stimulating the cGAS-STING pathway and amplifying the production of type I interferons. PARPi also improves anti-tumor immunity independently of BRCA background, as shown in vivo in C57BL/6 mice with MB49 xenograft tumors [142], but olaparib in conjunction with STING agonists impedes the growth of BRCA1-deficient breast cancer in wild-type mice but shows no impact in mice with STING-KO variants [143]. The action of PARPi in BRCA2-deficient tumors correlates well with immune-cell infiltration [144].

In a murine model of BRCA1-deficient ovarian cancer, the use of olaparib to inhibit PARP1 promotes antitumor immunity, especially when it is combined with PD-L1-blocking antibodies, and this action is STING-dependent in mice with tumor xenografts [145]. Notably, in many cancers with DDR defects, such as non-small-cell lung cancer (NSCLC) deficient in ERCC1 (excision repair cross-complementation group 1) or TNBC deficient in BRCA1, PARPi synergizes with IFN-γ to induce expression of programmed cell death ligand 1 (PD-L1) on the cell surface, thereby amplifying the effectiveness of immune checkpoint inhibitors (ICI) [146]. However, a contradictory observation has been reported in mouse and human ovarian cancer models, in which cancer cells exhibited divergent sensitivities to olaparib when analyzed in vitro compared to in vivo, with cells in the latter case showing a progressive development of resistance. This resistance may be tied to STAT3 activation in tumor cells in the case of PARPi-resistant BRCA1-deficient murine ovarian cancer [147].

### 5.4. Immune Checkpoint Blockade and cGAS-STING Pathway

ICB relies on the binding of specific monoclonal antibodies to inhibitory immune proteins PD-1, PD-L1, CTLA4 (cytotoxic T-lymphocyte associated protein 4), LAG3 (lymphocyte-activation gene 3). Here, the cGAS-STING pathway is also important [21]. Additionally, DDR and ICB hare tightly linked: deficiencies in DDR mechanisms can give rise to immunogenic cancer neoantigens, thereby amplifying the efficacy of ICB therapies [148].

## 6. Duality of the cGAS-STING Pathway Effects in Cancer

### 6.1. cGAS-STING Pathway Inhibition Promotes Carcinogenesis

Various known driver mutations significantly impact this pathway through protein-protein interaction networks. For example, it has been shown that oncogenic mutants of p53 binds TBK1 directly, prevent the formation of STING-IRF3-TBK1 complex. Specifically, p53R249S, p53R280K, and p53R248W mutations lead to impaired phosphorylation of TBK1 and its substrates (STING and IRF3) [110]. 

In contrast to autoimmunity [149], it is not known whether there are germline mutations in genes of the cGAS-STING pathway that may be important for primary carcinogenesis. Somatic perturbations in this pathway are widespread, and a fruitful field of research explores the epigenetic silencing of cGAS, STING, and functionally connected genes. Recently, for example, it has been demonstrated that hypermethylation of the promoters of these genes in human melanoma is an important factor in immune-cell evasion. Conversely, restoration of the normal cGAS-STING pathway boosts T-cell recognition of melanoma cells (a typical immunologically “cold” tumor) [150].

### 6.2. Tumorigenic Effects of the Excessive cGAS-STING Activity

As a rule, *nucleosomal* cGAS exerts a certain pro-oncogenic effect, as it inhibits HR (but not NHEJ) in the presence of double-stranded breaks through direct binding to PARP1 (via poly(ADP-ribose). It thus prevents binding to the replication fork-stabilizing protein Timeless. Genotoxic exposures such as etoposide, camptothecin and peroxide promote BLK (B-lymphoid tyrosine kinase), and the constitutive BLK phosphorylation of cGAS Tyr215 seems to retain cGAS in the cytoplasm. Dephosphorylation of cGAS (by unknown phosphatases) in response to DNA damage leads to its nuclear translocation. In many models (immortalized human HCA2-TERT fibroblasts, A549 lung-cancer cells, and LLC tumors in C57BL/6 mice) it is evident that cGAS knockout inhibits tumor growth [101]. This activity of cGAS presumably occurs independently of STING and also may promote growth of some tumors. 

It is well known that tumors whose initiation depends on BRCA1 loss or ATM have relatively high levels of inflammation and T-cell infiltration from the beginning of malignant growth, but these events do not lead to the elimination of early tumors due to activation STING and IFN genes. Thus, STING knockout results in decreased neoangiogenesis, increased CD8+ T-cell infiltration, and restoration of therapeutic resistance to double ICB [151]. It is not unequivocally proven, but it is quite likely, that nuclear cGAS can inhibit DNA repair, thereby contributing to oncogenesis at the level of the whole organism [101].

Prolonged activation and subsequent inflammation can lead to immune cell exhaustion, a phenomenon frequently observed in tumor environments. Although complete deactivation of the cGAS-STING pathway in primary tumors is rare, overstimulation of this pathway can paradoxically support cancer growth. This phenomenon is visible in activation markers such as TBK1 and IRF3 phosphorylation and heightened IFN-β and ISGs mRNA expression. Specifically, in TNBCs, overactive cGAS–IL-6–IL-6R signaling is linked to decreased immune-cell survival and increased cancer-cell growth [152]. Genotoxic stress in STING-positive cancers causes activation of this pathway, and some cells activate the non-canonical STING pathway, leading to abundant secretion of immunosuppressive IL-6 (this pathway also requires inhibition of ERK1/2 [153]. 

Chromosomal instability (CIN) in cancer cells manifests in tangible outcomes like aneuploidy. However, its root causes can vary significantly. For instance, 17 distinct CIN signatures have been identified, potentially reflecting 17 different etiologies [154]. Therefore, understanding the contexts that influence complex pathways such as cGAS-STING either in promoting or countering tumor progression is a formidable task. The inherent duality of the role of cGAS- STING in high-CIN cancers can be outlined as follows: cancer cells attract immune cells and die, but they simultaneously activate motility and distant metastasis [155]. 

In certain models of TNBC cells exhibiting high CIN, a reduction in cGAS–STING levels resulted in decreased IL-6 production and reduced survival of cancer cells. Indeed, the production of cytokines like IL-6 is regulated by non-canonical NF-κB signaling; moreover, this pathway likely disrupts conventional STING–TBK1–IFN signaling, thus suggesting that STING activation may have both anti- and pro-tumor effects, depending on transient states of the evolving tumor [75].

Additionally, the role of IFI16 is controversial: it may be involved in p53-mediated transcriptional activation by enhancing p53 sequence-specific DNA binding and modulating pP53 phosphorylation status. It may additionally be involved in the regulation of p53-mediated cell death, which also involves BRCA1. However, STING-mediated IFI16 degradation negatively regulates IFI16-mediated p53-dependent apoptosis in osteosarcoma and NSCLC cells [156].

## 7. Metastasis and cGAS-STING Pathway 

STING activation by cGAMP may curtail metastasis in specific models, such as the murine CT26 colorectal cancer, perhaps due to the inhibited suppression of T-cells by MDSC-driven ROS and nitric oxide synthase (NOS) [157]. In some other oncological contexts, STING activation is also extremely effective in inhibiting metastasis. For example, in lung adenocarcinoma (LUAD), migrated cancer cells may remain dormant for a considerable amount of time, escaping detection by NK and T cells. In some murine models, STING activation has been shown to be effective in suppressing the activation of such pre-disseminated cells [158].

Unfortunately, in high-CIN cancers, cGAS-STING promotes motility and metastasis [155,159], most likely due to the aforementioned connection between STING activation and upregulation of NF-κB, which subsequently elevates the expression of transcription factors associated with epithelial-mesenchymal transition (EMT). NF-κB activation is a serious pro-EMT factor (reviewed in [160,161]), concomitantly inducing various pro-inflammatory cytokines (such as IL-1, IL-6, TNFα, TGFβ). 

EMT in cancer is a complex and partially reversible phenomenon that mirrors developmental processes. It hinges on the transient adoption of migratory cell phenotypes that contribute to the spread of the primary tumor and seed metastases. Mechanistically, EMT is driven by the activation of a specific gene-expression regulatory signature that includes SNAIL, TWIST, SLUG, ZEB1, ZEB2, and E47 and complex switches in cytoskeletal proteins (from cytokeratin to vimentin, etc.); and adhesion proteins (from E-cadherin to N-cadherin etc.). Phenotypically, cells lose cell-cell junctions and apical-basal polarity and acquire fibroblast-like features such as filopodia and lamellipodia, which are important for high cell motility and invasiveness. 

There is a paradox at play: while the activation of pathways resembling antiviral defenses might be expected to curb cancer-cell growth and survival, high instability is actually associated with aggressive invasion and metastasis. Certain cancers, like breast and lung cancers, grow with induced cGAS-STING and simultaneous activation of the non-canonical NF-κB pathway, where the chain of events such as tumor necrosis factor receptor (TNFR)-dependent chronic activation of NF-κB-inducing kinase (NIK), followed by NIK-mediated p100 phosphorylation, p100 processing, and nuclear translocation of p52 and RELB, leads to changes in genes expression. These changes ultimately result in stimulated cancer-cell proliferation and neoangiogenesis, together with inhibition of apoptosis and robust induction of EMT [159].

STING promotes NF-kB activation and hence augmented transcription of its target genes, including TWIST, SLUG, and ZEB2 [162] (Figure 5). However, it is important to note that cGAS/STING activity alone is inadequate to induce EMT or account for invasive behavior; moreover, at least in some models, the activation of NF-κB has anti-tumor effects even when it occurs alongside aneuploidy (in nude mice with hepatic metastasis [163]). 

A recent study proposed a solution to the problem: it is CIN, rather than aneuploidy itself, that is a principal driver of cancer metastasis. In that case, chromosomes lagging due to CIN may become entrapped extranuclearly and are detected subsequently by cGAS-STING, which activates non-canonical NF-κB signaling. This pathway ultimately results in EMT and heightened cellular invasiveness, as shown in nude mice and ex vivo with patient-derived human breast-cancer cells [159].

Another metastasis-promoting role of STING activation is connected to tachyphylaxis, a diminished IFN responsiveness to repeated stimulation that occurs during persistent activation of cGAS-STING signaling, concomitant with non-canonical NF-kB pathway induction. In human TNBC, tumors with high GAS and low STING were associated with fewer tumor-infiltrating lymphocytes and were linked to decreased distant metastasis-free survival (DMFS), whereas tumors with low cGAS and high STING had characteristics associated with a more favorable prognosis [164].

In conclusion, the cGAS-STING pathway may promote metastasis if its activation is chronic. However, a simple answer to the question of how the cGAS-STING pathway contributes to EMT, metastasis, and interactions between tumor cells and their microenvironment remains largely elusive (reviewed recently in [165]), especially for treatment-resistant cancers. 

## 8. Pharmaceutical Achievements in the Regulation of the cGAS-STING Pathway 

The current focus is on synthetic STING agonists, which, unlike the natural secondary messenger cGAMP, are more potent and stable in vivo. For many tumors, STING agonists are the prime candidates for precision oncotherapy. In many mouse models, stimulating STING with small molecules has yielded promising outcomes [134,166,167,168]. This activation leads to a decrease in tumor size, primarily due to increased T-cell infiltration, tumor vesicle disruption, and heightened apoptosis. More importantly, it enhances the immunogenic properties of tumors that were previously resistant to immunological interventions, reverting resistance to anti-PD-1 therapies. The resulting secreted IFN type I has multiple effects: it activates antigen-presenting cells, including DCs; it stimulates natural killer cells (NKs); it hinders the mobilization of M2 macrophages and MDSCs; and it primes cytotoxic cells. Simple intratumoral administration of cGAMP works to some extent in mice with cancer cells such as T26, B16F10, and mSCC1 cells [169,170]. The early results, for example, with pancreatic cancer in mice, have demonstrated that cGAMP acts through STING T-cell-independent and TNFα-dependent hemorrhagic necrosis [134]. 

It should be noted that the tumor suppressor p53 and STING pathways are intimately intertwined (Figure 1 and Figure 2, left panels). Unlike the wild-type p53 protein, its mutant forms suppress the activity of cGAS and hence attenuate cGAS/STING activation [171]. As a result of post-translational modifications induced by DNA damage, p53 activates the expression of its target pro-apoptotic genes (Bax, Puma, Noxa) [172,173], thereby sensitizing cells to cGAS/STING-induced apoptosis [109,174]. In this respect, several publications have reported that MDM2 inhibitors that stabilize p53 [175,176]), including nutlin-3A, can also enhance the effect of cGAS/STING [109,177].

STING agonists and antagonists (Figure 6) have been extensively reviewed [178,179,180,181,182,183,184], so here we only emphasize that the development of these small molecules is challenging by the relatively large size of the 700-Da binding pocket in the STING molecule and differences between mouse and human STINGs. Also, it is difficult to minimize side effects such as excessive stress in T cells, which can lead to cytokine storms, apoptosis of cancer-associated cells instead of IFN secretion, or induction of the tolerogenic enzyme IDO (indoleamine-2,3-dioxygenase). There are other crucial points to consider, starting from the vast inter-species differences. For instance, 5,6-dimethylxanthenone 4-acetic acid (DMXAA) binds efficiently to mouse STING but not to human STING, leading to stark differences between preclinical [166,167,168] and clinical results [185] (few of them were published; reviewed in [19,186]). Desired effects might also be muted due to interference from other pathways. For instance, to achieve the intended effects of cGAMP or other STING agonists, it might be necessary to block pathways like JAK2-STAT3 in mice [187].

Compound 53 (Figure 6) revealed an extremely interesting feature of STING because it binds STING in its transmembrane area, inhibiting ion-channeling activity, unlike the STING agonists MSA-2 [188] and diABZl [68]. This binding leads to a marked increase in Golgi pH due to its intrinsic role as a proton channel. The pH shift is vital for the activation of the NLRP3 inflammasome and subsequent pyroptosis, results in IL-1β secretion, and is independent of lipidation and autophagy. It is well known that tumors frequently have pH gradients rather different from those of normal tissues; moreover, Golgi acidification may be important for cytoplasmic pH maintenance, at least in some cancer cells [189]. Therefore, any future studies on cGAS-STING oncological functions should pay attention to this previously unexpected mechanism.

Stabilized cyclic dinucleotide compounds, which activate human STING, are in clinical trials for potential application in cancer immunotherapy (Table 1). However, the toxicity limit of these agents may be tied to the initiation of necroptosis due to STING hyperactivation of a “shock-like” condition, a simultaneous increase in the levels of TNF and of various inflammatory cytokines. Nonetheless, clinical trials reported that cGAMP-similar CDN analogs like ADU-S100 monotherapies are not effective either in oncohematology or in therapy for solid tumors in patients [190,191,192]. Moreover, only slightly higher efficiency was found when this treatment was combined with anti-PD-1 antibodies (such as pembrolizumab), reviewed in 2022 by Samson and Ablasser [19] and in 2023 by Seok et al. [193]. 

The recently discovered Ulevostinag/MK-1454 [194] demonstrated the importance of phosphorothioate synthesis stereoselectivity [195,196]. The results of the NCT04220866 study are promising. This study compared two patient groups: one treated with Ulevostinag in combination with pembrolizumab, and the other treated with pembrolizumab alone, consisting of 8 and 10 patients, respectively. While adverse events were comparable between the two groups, the objective response rate (ORR) was significantly higher in the combination therapy group (50%) compared to the pembrolizumab-only group (10%). Progression-free survival (PFS) rates were 6.4 months for the combination-therapy group versus 1.5 months for the control group. Most importantly, there may be an increase even in overall survival (OS) after 913 days of follow-up.

The normal action of type 1 interferons in the body is local. Systemic administration of IFN is accompanied by a variety of side effects, the aggregate of which represents a “cytokine storm” and leads to imbalance in the cGAS-STING pathway, which harms the organism as a whole and compromises the effectiveness of antitumor therapy. Therefore, most preclinical and clinical studies of STING agonists involve intratumoral administration of the drug [NCT05549804, NCT04144140,]. MSA-2 is potentially well-suited for systemic oral administration, as its inactive monomeric form can penetrate into cancer cells more effectively than its active dimeric form. This monomer-dimer transition is pH-dependent; in acidic environments, MSA-2 exists predominantly as a monomer, thereby facilitating its preferential penetration into cells within tumor nodules, which typically have a lower pH. Once inside the cytoplasm, where the pH is neutral, MSA-2 transitions into its active dimeric form [188].

Novel methods for targeted delivery of STING agonists into tumors are in active development. This targeted approach can be achieved through transport molecules like immunoconjugates. Drug-antibody conjugates, sometimes referred to as ISACs (immune-stimulating antibody-conjugates), offer a promising option for precision cancer treatment. For instance, TAK-500 (TAK-676 conjugated to an anti-CRCC2 antibody targeting the cysteine-cysteine chemokine receptor 2), aims to block immunosuppressive intratumoral cells [197]. Clinical trials for TAK-500 commenced in 2022 [NCT05070247]. Another example is XMT-2056, an antibody-drug conjugate with a STING-agonist (di-ABZl) that binds a novel epitope of HER2 [17]. 

Some approaches that may seem unconventional yet hold potential. For instance, SYNB1891 is a probiotic containing live *Escherichia coli* designed to synthesize cyclic dinucleotides in situ under hypoxic conditions [198]. Another emerging field is the activation of STING to enhance the effectiveness of mRNA-based vaccines [199].

Despite these advancements, challenges and limitations remain, including low efficacy and adverse effects. While STING agonists in clinical trials generally are associated with mild adverse effects such as pyrexia, injection site pain, and diarrhea [193], new side effects like kidney dysfunction were observed in C57BL/6 mice following STING activation. These findings should prompt caution in human trials [200].

Recent studies revealed an unexpected role of bivalent ions (Mn^2+^) in the activation of the cGAS-STING pathway [201]. Administration of Mn^2+^ to cancer cells augmented their sensitivity to the presence of dsDNA by several mechanisms. First, the ions activated cGAS enzymatic activity, facilitating the production of the secondary messenger cyclic GMP-AMP (cGAMP). In addition, Mn^2+^ was shown to enhance STING activity by augmenting cGAMP/STING binding affinity [202]. Importantly, the same report claimed that Mn^2+^ itself may serve as a potent cGAS activator independently of the presence of dsDNA, hence forcing cancer cells to produce type I IFNs [202]. Thus, it is likely that Mn^2+^ has complex effects on the cGAS/STING pathway [203]. 

It should be noted however, that most of these studies have been carried out using xenografts of allogenic and syngeneic tumor cell lines (4T1 breast cancer, B16-F10 melanoma, pancreatic Pan02 [204], and colon CT26 [203]). This issue is important because Mn^2+^ can easily penetrate the BBB and, at high concentrations, can cause severe intoxication. It will be interesting to learn whether therapeutically significant amounts of Mn^2+^ are safe for use in humans. 

Cytoplasmic DNA accumulation can be augmented by inhibition of DNases such as TREX1. There are no clinical trials of TREX1 inhibitors yet, but their antitumor activity has already been shown [205].

### 8.1. Administration Schedules

The primary challenges and limitations arise from the need to guarantee localized drug action. Additionally, when targeting the cGAS-STING pathway, careful consideration is required in selecting the therapeutic regimen. This caution is needed because both prolonged activation and inadequate activation of the cGAS-STING pathway can result in unwanted chronic inflammation, which may promote tumor survival and metastasis.

It was suggested quite early that anti-tumor STING activation in oncotherapies should be “acute and moderate” and not “persistent or extensive” [19]. This assertion is now fully accepted: hit-and-run approaches are now preferred over prolonged treatments [206]. These ideas are in a sharp contrast to the experimentation with cycling regimes in combination with PARPi and other drugs [207]. 

### 8.2. STING Antagonists

Apparently, periodic inhibition and activation of the cGAS-STING pathway is also worth trying. Indeed, many small-molecule STING inhibitors have been developed, including irreversible ones [208] and heterobifunctional molecules designed as proteolysis-targeting chimeras [209]. Although these inhibitors were initially developed to treat autoimmune diseases, their potential for applications in oncology should not be overlooked. This consideration is particularly pertinent given the steadily increasing variety of available inhibitors in this category [210].

### 8.3. Nanosystems for Efficient Delivery of cGAS-STING Modulators

cGAMP and analogs work much better when they are aided by smart delivery vehicles. Since there are excellent recent reviews on this subject [211,212], we mention only that among recent breakthroughs and improvements using these approaches, the most interesting include polymerosomes made with matryoshka layers, CDN encapsulated with polyethylenimine (PEI), biodegradable disulfide cross-lined with polycarbonate and covered with polyethylene glycol (PEG) [213] so that it should unfold in the cytoplasm, polymerosomes [214], and stable cyclic dinucleotide nanoparticles with a hydrophobic nucleotide lipid (3′,5′-di(oleoyl-deoxycytidine) [215]. 

Optimization of liposome composition allowed researchers to obtain CDNs stably incorporated into the liposomes, protect CDNs from spontaneous degradation in the blood, and retain the activity of ADU-100 [216], and diABZI [217] and various CDNs in extracellular vesicles [218]. Moreover, virus-like particles containing long DNA strands encapsulated in cationic liposomes showed some promise in inducing the liquid-phase condensation of cGAS, thereby activating STING [219].

Some nanocomposites are believed to act through damage to the mitochondrial membrane concomitant with the release of mitochondrial DNA (mtDNA) and activation of cGAS-STING [220]. These ultra-small Cu2−XSe (CS) nanoparticles on the organosilicon nanoparticles are covered with cancer-cell membranes for efficient delivery.

### 8.4. STING as a Drug

Here, STING mutants such as N153S (first known from patients with the autoimmune disease STING-associated vasculopathy with onset in infancy) are especially valuable, as this mutant is hyperactive. In MC38 colorectal cancer cells, even an admixture of cells with this STING mutant greatly increased the sensitivity of the whole population of cancer cells to ICB in vivo, achieving sensitization of “cold” tumors via an increase in extracellular signaling, for example, in C57BL/6 mice with MC38 adenocarcinoma [221]. This important discovery paves the way for studies on STING variants as transgenes in anti-cancer gene therapy in vehicles such as adeno-associated viruses (AAV), oncolytic viruses (OVs) and others. Successful transduction of even a small proportion of cancer cells may elicit the ultimate anti-tumor effect. 

Also, using nanoparticle delivery, the recombinant TM-truncated STING protein, in combination with cGAMP, has been shown to effectively activate STING signaling both in vitro and in vivo [222]. 

## 9. Agonists of cGAS- STING Pathway in Distinctive Cancer Types and Perspectives of Personalization

It is important to consider the origin of the primary tumor, as well as tissues harboring distant metastases, to better understand the mechanism of progression for a particular tumor. Depending on the abundance of effectors or modulators of the STING pathway, the latter pathway may be regulated differently in different tissues. For example, cGAS is abundant in leukocytes and macrophages but has low expression levels in the brain, liver, prostate, skeletal muscle, skin, intestine, testes, and stomach. Thus, many of the described effects are pertinent for cGAS in infiltrating blood cells. STING is abundant in endothelial cells and cardiomyocytes, macrophages, and DCs. Consequently, the same stimulus (e.g., DNA damage) may have different impacts on STING pathway activation in different tissues. These simple facts are important when considering different modulators of the STING pathway. Therefore, the therapeutic implications of cGAS-STING pathway modulation should be considered primarily in the context of the interactions between cancer and immune cells.

### 9.1. Gliomas 

Reflecting on the various mechanisms of modulating the cGAS-STING pathway in oncology, it is important to realize that in many tissues, the level of STING expression is low, especially in most normal brain cells and human glioblastoma cells (a classic example of an immunologically cold tumor). In contrast, it appears that blood vessels associated with glioblastoma are STING-positive, so they may respond to STING agonists, such as ADU-S100, with secretion of inflammatory cytokines. At least in some mouse models of glioblastoma (especially in GL261 cells, where both KRAS (Kirsten RAt Sarcoma viral oncogene) and p53 are mutated, ADU-S100 causes massive infiltration by inflammatory macrophages, neutrophils, and NK, prolonging survival [223]. It is possible that STING can be activated in human glioblastoma, where it is strongly expressed in tumor-associated blood vessels [223]. 

In other gliomas, STING may also be important despite relatively low expression levels. For example, STING may increase resistance to temozolomide (TMZ) [224]. 

IR can also induce secretion of damage-associated molecular patterns (DAMPs) (such as ATP and HMGB1). DAMP levels in H3.3-G34R and H3.3-WT mouse and human pHGG cells in response to IR were also reduced by these inhibitors. Therefore, the cGAS/STING pathway affects stimulation of DAMPs released upon IR-induced DNA damage in mice [224].

The PARP inhibitor pamiparib appears to penetrate the blood–brain barrier (BBB), and a STING agonist (diABZl) increases the efficacy of treatment by checkpoint CHK1/2 inhibition (together with 3.3-G34R pHGG and AZD7762) in a murine model [225].

### 9.2. Pancreatic Ductal Adenocarcinoma (PDAC)

It is important to emphasize again that STING-positive cancers tend to be immunologically “hot”, while STING-negative cancers tend to be “cold”, with the clear exception of highly aggressive pancreatic ductal adenocarcinoma (PDAC) among other cold cancers, as they do not fall into the dichotomous “temperature” classification [226]. This type of cancer often exhibits fibrosis/desmoplasia, which is characterized by infiltration of pancreatic stellate cells, fibroblasts, and immune cells, all contributing to a complex extracellular matrix. Interestingly, STING expression is prevalent in 90% of PDAC cases, whereas tumors expressing both cGAS and STING are associated with somewhat improved patient survival. Such double-positive tumors have strong infiltration by cytotoxic CD8+ cells [227]. This infiltration is presumably due to activation of the canonical cGAS-STING pathway. However, in PDAC the STING-PERK-eIF2α axis is also important because it promotes fibrosis [228].

In PDAC, CD73 is emerging as a promising therapeutic target. Its presence is linked with poorer patient survival. CD73, expressed on both TAMs and PDAC cells, influences myeloid cells to adopt an M2-like phenotype. These M2-like cells, when in proximity to PDACs, promote tumor growth and diminish the genotoxic stress induced by treatments like ionizing radiation (IR) or gemcitabine. Intriguingly, the absence of CD73 appears to stimulate the cGAS-STING pathway [229]. 

In PDAC cells with p53 mutations, the formation of micronuclei is linked to the mitotic phosphorylation of SUMO-specific protease 3 (SENP3). p53 activates SENP3, which in turn promotes cellular senescence, which correlates with longer survival outcomes for PDAC patients. Furthermore, SENP3 activity enhances anti-tumor immune responses, a mechanism that is facilitated by cGAS in mice [230]. 

### 9.3. Melanoma

In melanoma research, the B16F10 murine syngeneic model, a classical “cold” cancer, is popular and it is known to respond to STING agonists [231]. For immune-checkpoint anti-tumor therapies, the cGAS-STING pathway is crucial. In experiments, mice lacking cGAS exhibited reduced B16 melanoma growth when they were treated with a PD-L1 antibody. In line with this, direct cGAMP delivery into the muscles hindered melanoma growth and prolonged the survival of the tumor-bearing mice. Even though cGAMP should be rapidly degraded by the enzyme, it might not have sufficient time to exert its effect in murine models in vivo [232]. 

### 9.4. Oncohematology

The potential effects or the interferon-independent actions of STING might be especially useful in oncohematology. For instance, a specific STING agonist, DMXAA, induces the expression of IFN-β and various inflammatory cytokines. This expression not only promotes maturation of dendritic cells, but also results in a remarkable proliferation of leukemia-specific T cells, extending survival in two acute myeloid leukemia (AML) murine models [166]. 

Of special note, DDX41 plays a significant role in myeloid malignancies, including acute myeloid leukemia (AML) and myelodysplastic syndromes (MDS). Mutations in DDX41 are found in approximately 2–5% of AML and MDS patients, highlighting its importance in these conditions [233]. 

### 9.5. Personalization on the Basis of Genomic Profiling

Importantly, STING polymorphisms exist in the human population: R71H-G230A-R293Q (HAQ) in 20.4% of those surveyed, R232H in 13.7%, G230A-R293Q (AQ) in 5.2%, and R293Q in 1.5%. The R71H substitution has a notably lower intrinsic activity; the R232H and R293Q variants are poorly responsive to bacterial CDNs such as c-di-AMP and 3′,3′-cGAMP, and these variants are important for bacterial infections [234,235]. Therefore, personalized approaches [236] should be accordingly considered in oncology, as proposed by Gogoi et al. in 2020 [191]. 

### 9.6. Modulating cGAS-STING According to the Tumor Peculiarities

Personalized treatments and individual variances remain an important avenues of research [19]. Certain STING agonists appear to operate independently of T cells, despite the fact that STING agonist-triggered type 1 interferon affects a variety of cells, including T-lymphocytes. Therefore, their tumor-reducing effects ought to be precisely calibrated to optimize their therapeutic effect [237] on innate signaling via antigen-presenting cells (APCs) in the tumor microenvironment (TME). This pathway thus facilitates cross-priming of tumor antigen-specific CD8+ T cells.

Recent studies have revealed that ARID1A (AT-rich interaction domain 1A), an integral part of the highly conserved SWItch/Sucrose Non-Fermentable (SWI/SNF) chromatin-remodeling complex, facilitates RNF8-driven ubiquitination and subsequent degradation of Chk2. In cancers, ARID1A deletion promotes STING-mediated innate immune responses. Consequently, tumors characterized by mutations or reduced expression of both ARID1A and ATM/Chk2 exhibited increased lymphocyte infiltration and were less aggressive (in C57BL/6 mice with ID8 cells) [238]. This observation aligns with the recognized synergy between lymphocyte mobilization and ATM deficiency [239].

For cancers with KRAS mutations, it is established that mtDNA release into the cytoplasm can activate STING, and suppression of STING has been linked to immune evasion in lung-cancer cells with KRAS mutations (in vitro and in SCID mice with xenografted human lung adenocarcinoma A549 cells) [240].

Epidermal growth factor receptor (EGFR) enhances STING activity through phosphorylation (in vitro and in mice) [241]. Therefore, evaluating the influence of all recognized hyperactive EGFR mutations, especially those integral to oncogenesis, on STING activation, and degradation becomes paramount. Additionally, the assessment of pharmacologically relevant EGFR inhibitors is warranted.

In HER2+ breast carcinoma, it has been demonstrated that HER2 (unlike EGFR) suppresses cGAS-STING signaling, hindering the cancer cells from producing cytokines, entering senescence, and undergoing apoptosis. This process occurs through the recruitment of AKT1 to phosphorylate TBK1, thereby preventing TBK1-STING interaction. This understanding opens up a new approach to using STING agonists, suggesting that they should be combined with inhibitors of the HER2-AKT1 cascade [242]. Furthermore, recent findings indicate the particular importance of IFI16-STING in HER2+ breast cancer: IFI16-dependent STING activation is mechanistically important in anti-HER2 responses, and the epigenetic suppression of IFI16 is linked to resistance against anti-HER2 antibodies [243].

In personalizing oncologic treatment and deciding whether to incorporate STING agonists, it is advisable to test tumors for high chromosomal instability (CIN). Accordingly, the careful selection of STING modulators is essential. However, directly measuring CIN in a practical setting presents challenges. Therefore, surrogate markers like aneuploidy should be routinely assessed in tumor biopsies to guide these therapeutic choices. Moreover, considering the problem of the decline in IFN response with continued STING activation, it has been proposed that a subset of patients might benefit from inhibiting the cGAS-STING pathway to control tumor –induced chronic inflammation and the resulting immune suppression (characterized by high cGAS and low STING levels) [164].

## 10. Perspectives for Efficient Synergies between cGAS-STING Regulation and Other Anti-Cancer Therapeutic Modalities

### 10.1. Interactions between Different Anti-Cancer Drugs

Complex combinations of STING agonists with diverse drugs should be explored (Figure 7). In TNBC contexts, for example, STING agonists improve outcomes of IL-2 and anti–PD-1 checkpoint blockade therapies (involving IL2 and anti-PD-1) in TNBC animal models. In particular, STING agonists amplify NK activation—a critical factor in metastasis inhibition in murine models [244]. To enhance the specificity of their action, the focus should be on how STING can be delivered exclusively to targeted cells.

Combination of a systemic STING agonist, MSA-2 (Figure 6), with osimertinib, a third-generation EGFR inhibitor, leads to a decline in advanced pulmonary embolism, perhaps through the action of STING on the tumor microenvironment, in mice [245]. 

Broad protein-kinase inhibitors and autophagy inducers such as sorafenib promote degradation of cGAS, TBK1, and IRF3, thus inhibiting the recruitment of STING with TBK1 and IRF [246]. Such compounds may find future use in many cancers marked by excessive STING activation, not only in renal and hepatocellular carcinomas, when properly combined with other drugs. 

In conjunction with certain cytotoxic therapies, checkpoint kinase 1 (Chk1) inhibition modifies the internal cellular environment but sometimes suppresses expected responses [247]. One proposed solution is to supplement Chk1 inhibitors with STING agonists [248].

The challenges of designing mechanism-based drugs are exemplified by the complexity and unpredictability of drug interactions. For instance, palbociclib, an inhibitor of the cyclin-dependent kinases CDK4 and CDK6 used in HR+/HER2- cancer treatment, has been observed to directly bind to and inhibit STING [249]. Another example: aspirin can directly acetylate cGAS, thereby inhibiting its function [250]. 

Moreover, the interplay between different DNA sensors is remarkable. For instance, elevated Z-form mtDNA, increased ZBP1 expression, and IFN-I signaling were recently discovered in cardiomyocytes exposed to doxorubicin, which is notoriously known to be cardiotoxic in some patients. ZBP1 is a cooperative partner for cGAS that sustains IFN-I responses to mtDNA instability and leakage [251]. Therefore, the control of side effects in IFN-inducing therapies should be explored through various mechanistic routes.

### 10.2. Cell-Based Therapies 

While the inherent T cell receptors (TCRs) usually demonstrate low affinity for self or tumor antigens, a patient’s lymphocytes may be reprogrammed with chimeric antigen receptors (CARs). However, challenges persist: solid tumors emit signals that deter T cells and hide specific oncomarkers [252]. STING agonists may be an ideal match with CAR-T cells, and intensive studies are being carried out to validate this combination. Indeed, when combined with the STING agonists DMXAA or cGAMP, CAR-T cells significantly improved tumor management, perhaps due to a prolonged presence of CAR T cells within the tumor microenvironment in mice [252,253].

### 10.3. Immunotherapies Such as Oncolytic Viruses (OV)

The initial concept behind the use of oncolytic viruses against cancer cells was rooted in the ability of many viruses to selectively replicate within tumors, subsequently killing the tumor cells. However, a more contemporary perspective posits that OVs function as a form of immunotherapy. This mechanism is often termed “immunogenic cell death”, wherein virus-infected tumor cells, while dying, release DAMPs that attract immune cells and stimulate antitumor responses. 

From a mechanistic perspective, a compromised STING pathway increases the sensitivity of cancer cells to direct viral oncolysis. This change occurs because such impairment leaves the cells defenseless against DNA viruses and retroviruses. This condition is seen in Talimogene laherparepvec (T-VEC, GM-CSF bearing HSV) in melanoma [254], in HSV-based OVs in ovarian cancer cells [255], and in mouse colorectal carcinoma [93]. Therefore, OVs are prime candidates for modern oncotherapies against STING pathway-impaired cancers. Moreover, STING-agonistic sequences such as c-di-AMP producing *disA* were used as transgenes in OVs [256], and this approach was also suggested independently [257]. 

Among poxviruses, engineered vaccinia virus has been extensively investigated as an OV for aggressive cancers. For example, genetic engineering yielded a potent immune-activating variant of MVA (rMVA, MVA∆E5R-Flt3L-OX40L). Removing a specific gene and incorporating two others resulted in a powerful anti-tumor response that was contingent on several cellular pathways [258].

Interestingly, killed virus in some settings produces excellent anti-cancer effects, underlining immuno-activation as the predominant mechanism of action of at least some OVs. For example, inactivated modified vaccinia virus in human or murine plasmacytoid dendritic cells (pDCs) induces type I IFN, whereas live vaccinia virus fails. Here, viral DNAs in the cytosol activated cGAS and downstream activated STING, IRF3, IRF7, IFN gene expression, and tumor death through the generation of antitumor CD8+ and CD4+ effectors in murine models [259].

It is also important to remember that the persistent activation of STING may impede the action of OVs (especially of OVs with dsDNA genomes such as poxviruses, herpesviruses, and adenoviruses) in cancer cells both in vitro and in vivo [260]. Therefore, temporary cGAS-STING inhibition is an option to explore, but this exploration would require extensive design of new small molecules. Conversely, the adverse effects of OVs may be controlled via various antiviral substances, as is done with new HIV-1 biologicals (to give an example [261]).

## 11. Conclusions

As a result of the incessant struggle of multicellular organisms against infections, the evolution of innate immunity has shaped the cGAS-STING pathway to have rather complex relationships with aging and oncogenesis. While this anti-pathogen sensor adeptly detects malignant cells with abnormal nuclei [103], its functions in the aging organism have become so convoluted that some cancers distort its function and use it against the immune system [99,164]. Therefore, additional studies are required to properly employ modulators of this pathway in oncology, especially in light of the discovery of the connection between the cGAS-STING and IL6-STAT pathways [152]. Additionally, it is crucial to highlight the challenges that arise from extrapolating results from animal models to humans. The limitations are exemplified by the fact that mouse dendritic cells do not produce IFN-λ1, a molecule prevalent in certain human cells [262]. Also, the adverse effects of STING agonists remain a challenging problem (reviewed recently [179]). 

Many questions about the cGAS-STING pathway remain unanswered or only partially resolved. For example, it is evident that cGAS differentiates between viral and host DNAs. However, the distinctions between self DNA in healthy and malignant cells are subtle. As a result, the cGAS-STING pathway may not be adequately activated during carcinogenesis. Therefore, significant research efforts are needed to find pharmacological approaches for desirable modulation of this pathway in health and disease.

## Figures and Tables

**Figure 1 pharmaceuticals-16-01675-f001:**
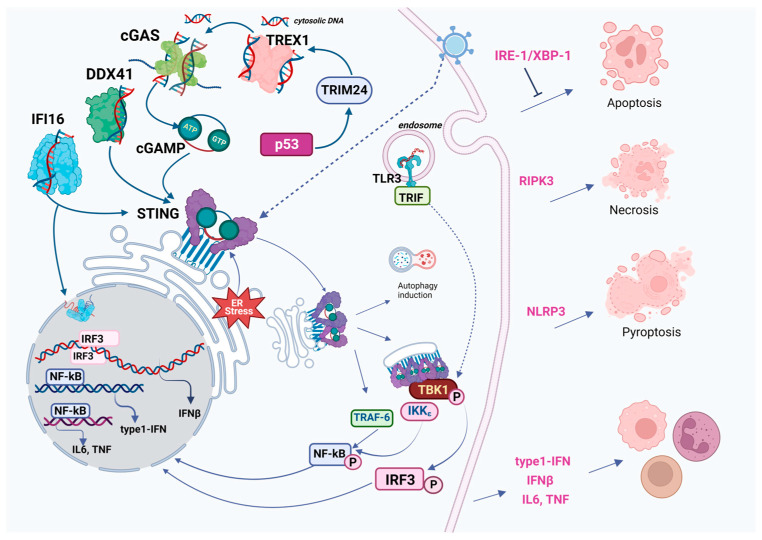
Paradigmatic outline of the cGAS-cGAMP-STING-TBK1-IRF3-IFN pathway. Note the 2:2 stoichiometry in the cGAS-DNA complex. Inputs from DNA sensors IFI16 and DDX41 are also important in some cells. Created with Biorender.com.

**Figure 2 pharmaceuticals-16-01675-f002:**
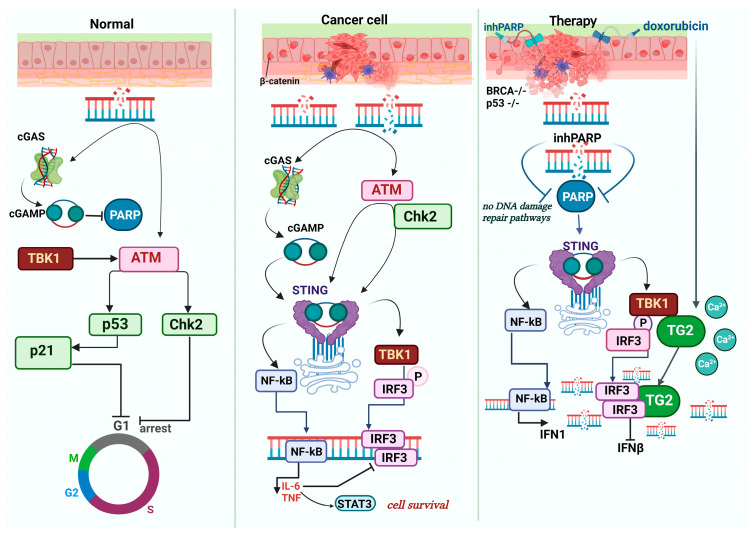
cGAS-STING in the prevention of carcinogenesis and its dualistic role in cancer progression. Created with Biorender.com.

**Figure 3 pharmaceuticals-16-01675-f003:**
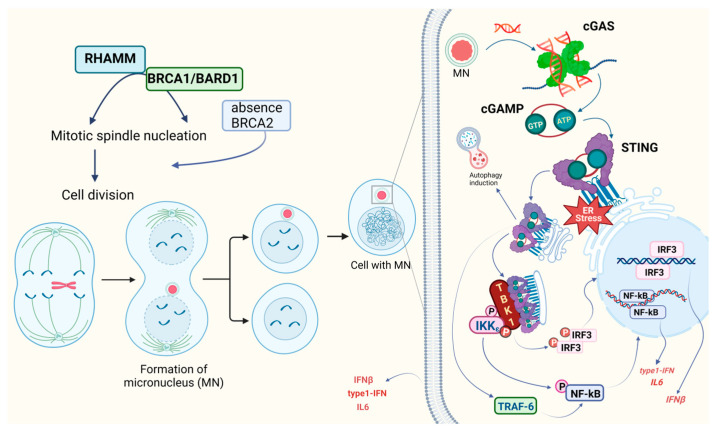
Micronucleus formation in cancer cells and subsequent activation of the cGAS-DNA pathway. Created with Biorender.com.

**Figure 4 pharmaceuticals-16-01675-f004:**
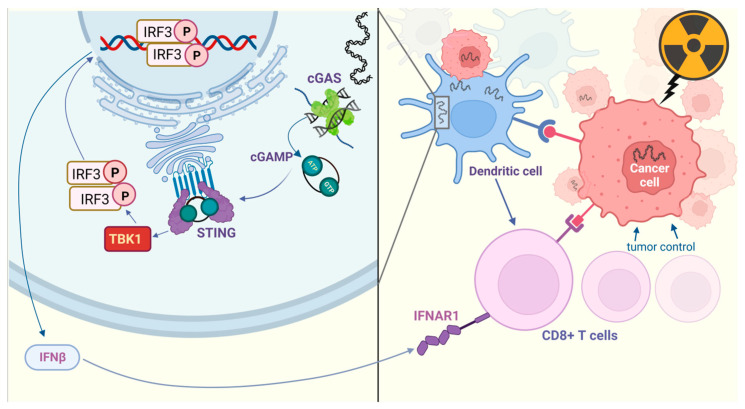
The role of the cGAS-STING pathway in tumor response to radiotherapy. Created with Biorender.com.

**Figure 5 pharmaceuticals-16-01675-f005:**
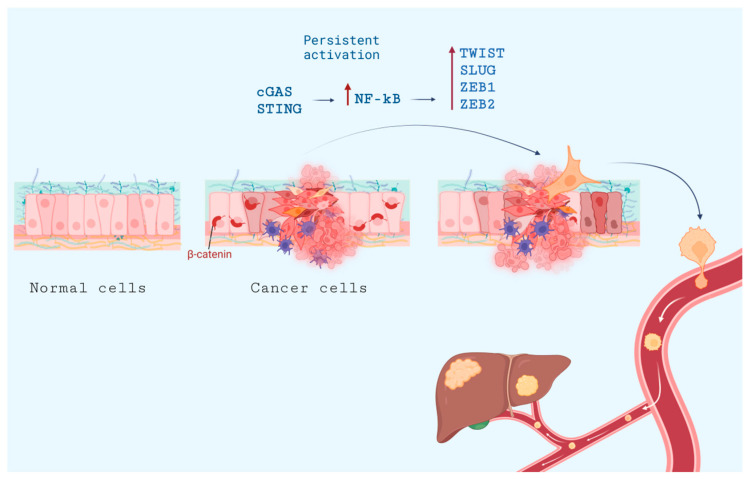
Outline of consequential primary tumorigenesis, epithelial-to-mesenchymal transition, and distant metastasis. Uncontrolled chronic activation of cGAS-STING may boost EMT and metastasis through NF-kB. Created with Biorender.com.

**Figure 6 pharmaceuticals-16-01675-f006:**
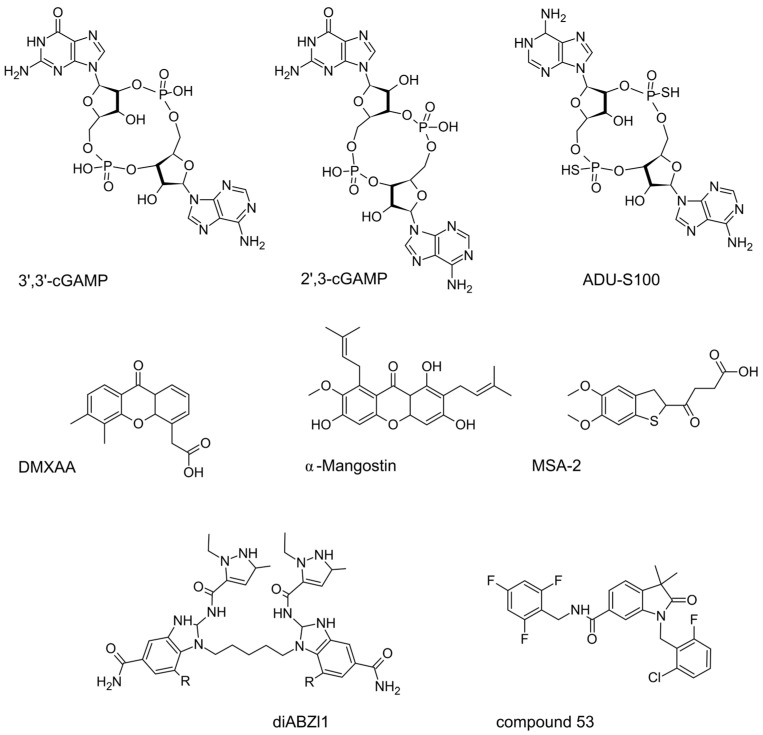
Structural formulae of the most well-studied cyclic dinucleotides and non-nucleotide STING agonists. ADU-S100 is especially efficient in murine cancer models like B16, CT26, and 4T1. Compound 53 works by a different mechanism. Created with ACD/Chemsketch.

**Figure 7 pharmaceuticals-16-01675-f007:**
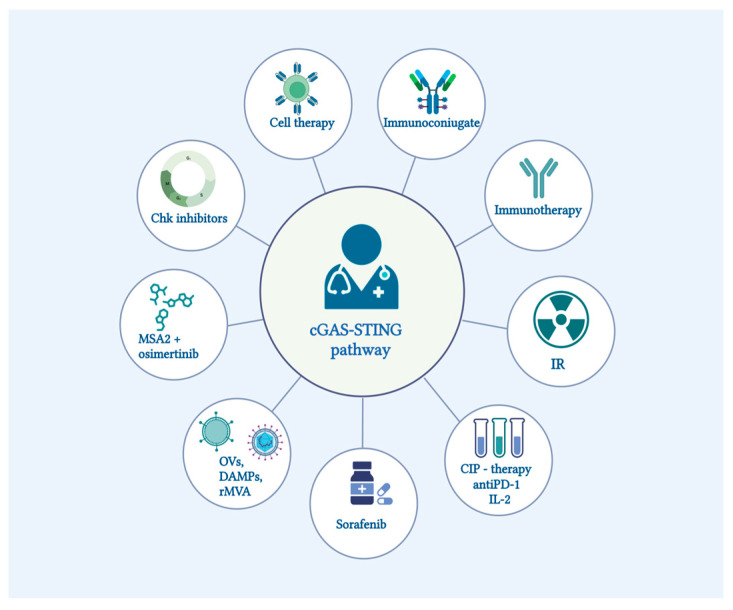
Diagram of numerous possibilities for synergistic applications of cGAS-STING agonists in oncotherapy. Created with Biorender.com.

**Table 1 pharmaceuticals-16-01675-t001:** Representative clinical trials of cGAS-STNG agonists in oncology.

Treatment Modality	Cancer Type	Compound	Status	Co-Treatment	Phase	Study ID
cGAS activation	var.	MnCl_2_	A	PD-1 ab	I	NCT03991559
	var.		A	radiotherapy	I/II	NCT04873440
	PanC		A	nab-PX, GC, PD-1 ab	I/II	NCT03989310
	OC		A	nab-PX, Pt, PD-1 ab	I/II	NCT03989336
	BTC		A	nab-PX, GC, PD-1 ab	I/II	NCT04004234
STING activation	var.	ONM-501	A	PD-1 ab	I	NCT06022029
	var.	BI 1387446	A	PD-1 ab	I	NCT04147234
	data	BMS-986301	A	PD-1 ab, ±CTLA-4 ab	I	NCT03956680
	HNC	ADU-S100	E	PD-1 ab	II	NCT03937141
	var.	ADU-S100	E	CTLA-4 ab	I	NCT02675439
	var.	ADU-S100	E	PD-1 ab	I	NCT03172936
	var.	E7766	C	N/A	I	NCT04144140
	NMIBC	E7766	E	N/A	I	NCT04109092
	var.	CDK-002/ExoSTING	C	N/A	I,II	NCT04592484
	var.	IMSA101	C	ICI	I,II	NCT04020185
	var.	IMSA101	A	radiotherapy, PD-1 ab	II	NCT05846659
	NSCLC, RCC	IMSA101	A	radiotherapy, PD-1 ab	II	NCT05846646
	var.	MK-1454	C	PD-1 ab	I	NCT03010176
	HNC	MK-1454	C	PD-1 ab	II	NCT04220866
	melanoma etc	SB11285	A	PD-1 ab	I	NCT04096638
	var.	GSK3745417	A	PD-1 ab	I	NCT03843359
	AML, HR-MDS	GSK3745417	A	N/A	I	NCT05424380
	var.	KL340399	A	N/A	I	NCT05549804
	var.	SNX281	A	PD-1 ab	I	NCT04609579
	var.	TAK-676	A	PD-1 ab, Pt, 5-FU	I,II	NCT04420884
	var.	TAK-676	A	radiotherapy, PD-1 ab	I	NCT04879849
	HNC	TAK-676	C	Pt, 5-FU, PX	I	NCT06062602
	sarcoma, MCC	CRD3874-SI	A	N/A	I	NCT06021626
STING activation by live bacteria	var.	SYNB1891	C	PD-L1	I	NCT04167137
Antibody-directed STING activation	var.	TAK-500	A	PD-1 ab	I	NCT05070247
	HER2+ cancers	XMT-2056	E	N/A	I	NCT05514717

AML—acute myeloid leukemia, BTC—biliary tract cancer, 5-FU—5-fluorouracil, ICI—PD1/PD-L1 and/or non-PD1/PD-L1 immune checkpoint inhibitors, HNC—head and neck cancer, HR-MDS—high-risk myelodysplastic syndrome, MCC—Merkel cell cancer, NSCLC—non-small cell lung cancer, NMIBC—non-muscle invasive bladder cancer, OC—ovarian cancer, PanC—pancreatic cancer, Pt—platinum-based chemotherapy, PX—paclitaxel, RCC—renal cell cancer, var.—various solid tumors and/or lymphomas. Status: A—active; C—completed; E—ended (suspended, terminated, withdrawn).

## Data Availability

Data sharing is not applicable.

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
