# Peer review of "At the Crossroads of the cGAS-cGAMP-STING Pathway and the DNA Damage Response: Implications for Cancer Progression and Treatment"

_pharmaceuticals, 2023, doi:10.3390/ph16121675_

Round 1

Reviewer 1 Report

Comments and Suggestions for Authors

The review article written by Korneenko et al., is well structured and of great importance to the readers. A minor comment I have for the authors would be to include in section 10, Perspectives for efficient …on line 743, an outline/diagram with the current drugs used for the various diseases as a take home message. Apart from that I believe that the article is updated and includes most of the current knowledge. Additionally, the authors could expand a bit more on IFNs and the antiviral contribution of this interesting pathway especially after the SARS-Cov-2 pandemic. 

Author Response

Please see the attachment (Rev 1).

Reviewer 2 Report

Comments and Suggestions for Authors

Great works have been present in this manuscript! This review systematically summarizes the significant role of cGAS-cGAMP-STING and DNA damage in cancer progression and treatment. This paper is worth reading and studying.

I think the article can be published in pharmaceuticals after some concerns that I list below are addressed:

1. Abstract

   There were two spelling forms of interferon: Line 19 IFN; and Line 23, interferon.

2. The sections (3, 4, 5, 7, 9, 10) should be supplied with some figures (such as figure 1 or 2 ). The attractive figures could help us understand the description related. 

3. The color of some words is red. (Line 332-363)

Author Response

Please see the attachment (Rev 2).

Reviewer 3 Report

Comments and Suggestions for Authors

I have carefully reviewed the article titled "At the cross-road of cGAS-cGAMP-STING pathway and DNA damage response: implications for cancer progression and treatment" by Tatyana V. Korneenko et al. I appreciate the authors' efforts in discussing the role of the cGAS-cGAMP-STING pathway in cancer progression and the development of new treatment modalities. Overall, the article provides important insights into the complex interplay between DNA damage response and the cGAS-cGAMP-STING pathway in the context of cancer.

Major Comments:

Clarity of the Introduction: The introduction provides a good overview of the cGAS-cGAMP-STING pathway; however, it would be helpful to provide more clarity on the specific roles of each component in the pathway and their molecular mechanisms.

Organization and Flow: The article lacks a clear structure, making it challenging for readers to follow the logical progression of the content. I recommend reorganizing the sections to improve the overall flow and coherence of the article.

Lack of In-text Citations: Several statements are made throughout the article without appropriate in-text citations to support the claims. It is essential to provide references for all relevant information and claims to ensure scientific accuracy and credibility.

Insufficient Discussion on Treatment Modalities: The article briefly mentions the development of new treatment modalities related to the cGAS-cGAMP-STING pathway but does not delve into specific therapeutic approaches or strategies. I suggest expanding this section to provide a more comprehensive discussion on potential treatment modalities and their implications.

Minor Comments:

Abbreviations: It would be helpful to provide a list of abbreviations used in the article, especially for less commonly known terms such as BRCA and PAMP.

Fig. 1: The article refers to Figure 1 several times, but Figure 1 is not included in the manuscript. Please ensure that all figures and their corresponding captions are included.

Questions for Clarification:

How does the cGAS-cGAMP-STING pathway contribute to the epithelial-to-mesenchymal transition (EMT) in treatment-resistant cancers?

Are there specific DNA damage patterns or characteristics that preferentially activate the non-canonical nuclear factor κB transcriptional response?

Can you provide examples of specific interferon-stimulated genes (ISGs) that are regulated by the cGAS-cGAMP-STING pathway in the context of cancer?

How does the cGAS-cGAMP-STING pathway interact with other immune signaling pathways, such as the NF-kB pathway, in the context of cancer progression?

Are there any known genetic or epigenetic alterations that can dysregulate the cGAS-cGAMP-STING pathway in cancer cells?

What are the potential therapeutic approaches targeting the cGAS-cGAMP-STING pathway in the treatment of cancer?

Can you elaborate on the role of proton channels in the cGAS-cGAMP-STING pathway and their significance in cancer progression?

Are there any clinical trials or ongoing research investigating the modulation of the cGAS-cGAMP-STING pathway as a therapeutic strategy for cancer treatment?

How does the cGAS-cGAMP-STING pathway influence the metastatic potential of cancer cells?

What are the challenges and limitations in targeting the cGAS-cGAMP-STING pathway for cancer therapy, and how can these be overcome?

In conclusion, the article provides valuable insights into the intricate relationship between the cGAS-cGAMP-STING pathway and DNA damage response in cancer. However, several improvements are needed to enhance the clarity, organization, and scientific rigor of the manuscript. Addressing the major and minor comments outlined above will significantly strengthen the article.

Based on the current state of the manuscript, I recommend major revisions before considering it for publication. The revised version should address the comments and suggestions provided in this review. Once 

Comments on the Quality of English Language

The manuscript would benefit from a thorough proofreading to correct grammatical errors and improve the clarity of the language used.

Author Response

Please see the attachment (Rev 3).

Reviewer 4 Report

Comments and Suggestions for Authors

Title; At the cross-road of cGAS-cGAMP-STING pathway and DNA damage response: implications for cancer progression and treatment

Comments;In my view, the results obtained in this study are worthy for publication. The manuscript needs major essential revision before publication. I would like to overview the revised version of the manuscript. I have the following comments/suggestions for authors to address before final decision on the manuscript.

1. The relationship between the cGAS-STING pathway and oncogenesis is mentioned, but it would be helpful to specify the different cancer types or scenarios where this pathway may play a role.

2. The phrase "non-canonical nuclear factor κB transcriptional response" is quite technical; therefore, authors should consider providing a simplified description for a broader audience.

3. The term "micronuclei" might need some explanation for readers who are not familiar with cell biology.

4. Mention the key highlights of the review article in the abstract. Also, mention the time duration from which the data is collected and presented.

5. The questions raised in the last paragraph of the Introduction section should be answered in the conclusions section.

6. The origin of the arrows showing different outcomes (Apoptosis, Necrosis, and Pyroptosis) is not clear in Figure 1.

7. In the Introduction section the author should refer to the research paper and comment on recent in-silico techniques. It will be good information for the readers. I would like to recommend several papers, among many others, providing further explanation on this topic: PMID: 

31133639 PMID: 31903852 PMID: 33465692 

8. The idea that this anti-pathogen sensor has been repurposed to detect abnormal nuclei in malignant cells is intriguing and underscores the pathway's versatility. However, it would be helpful to mention some specific examples or evidence of this repurposing.

9. The mention of cancer co-opting elements of the cGAS-STING pathway is a significant insight, but it would enhance the review to include some examples or mechanisms through which this co-opting occurs.

10. The call for additional studies to explore modulators of the cGAS-STING pathway in oncology is well-justified, but it might be valuable to suggest potential directions or areas where further research is particularly needed.

11. While the introduction mentions the "explosive growth" in articles on the topic, it lacks specific numbers or trends. Providing data or statistics on the growth of research in this area would enhance the statement's impact.

12. The section on various forms of nucleic acids and their potential recognition as foreign elements is valuable. Adding references to relevant studies or examples of how these forms are perceived by the immune system would enhance this portion.

13. The theoretical framework presented regarding the recognition of foreign and self-signals by the organism's friend-or-foe recognition system is insightful. It would be helpful to briefly mention the role of the cGAS-STING pathway in this context, connecting it to the central theme of the review.

14.Authors have suggested including the purpose and scope of the review article in the introduction to provide a clear roadmap for the readers.

15. Authors have advised to begin with a brief definition and background of the STING-cGAS pathway for readers who may not be familiar with it.

16. Define acronyms like IFN (Interferon) and EMT (Epithelial-to-Mesenchymal Transition) the first time they are mentioned to aid understanding.

17. Provide a concise overview of the relevant biological mechanisms before delving into the complex context-dependent roles of the pathway.

18. Discuss the potential therapeutic implications or practical applications of understanding the pathway's roles in cancer treatment.

19. Authors have suggested citing primary sources and studies to support the claims, especially when discussing the pathway's roles in cancer progression.

20. Authors have to clearly distinguish between in vitro and in vivo findings to highlight the relevance to real-world cancer scenarios.

21. Subheadings should be clear to organize the different aspects of the cGAS-STING pathway and its relationship with cancer.

22. Conclude the article with a concise summary of the key takeaways, highlighting the implications for future research and therapeutic development.

23. Use transitional sentences or phrases to smoothly connect one idea or section to the next. For example, when transitioning from the role of cGAS-STING in cancer to its interaction with BRCA1 and BRCA2, use a sentence that bridges the two concepts to maintain the text's flow.

24. Clearly explain the relevance of specific findings and observations. For example, when discussing STING's activation during H. pylori infection, clarify how this relates to cancer progression or other relevant outcomes.

25. For each cell death pathway, provide more details and specific findings related to STING activation and its role. Describe the mechanisms involved, key molecules, and their implications in different contexts, such as autoimmune conditions, cancers, or infections.

26. Discuss the significance of STING mutants, such as N153S, and their role in sensitizing "cold" tumors to immune checkpoint blockade (ICB). Explain how STING can be used as a transgene in various vehicles for therapeutic purposes.

27. Mention the potential toxicity and side effects associated with STING modulators and discuss strategies to minimize adverse effects. This will provide a more balanced view of their clinical application.

Comments on the Quality of English Language

Minor editing of English language required

Author Response

Please see the attachment (Rev 4).

Round 2

Reviewer 3 Report

Comments and Suggestions for Authors

Suggestions for Improvement:

Clarify the role of other DNA sensors: While the review focuses primarily on the cGAS-cGAMP-STING pathway, it would be beneficial to provide a more comprehensive overview of the other DNA sensors involved in the innate immune response. This would help readers understand the broader context and the relative importance of the cGAS pathway.

Address limitations and challenges: The manuscript briefly mentions the challenges in developing cGAS-STING modulating molecules and immunotherapies. It would be valuable to discuss these challenges in more detail, including the current limitations and potential strategies to overcome them.

Expand on clinical implications: The manuscript briefly mentions the implications of the cGAS-STING pathway in cancer treatment modalities. It would be beneficial to provide more insights into the potential clinical applications, such as ongoing clinical trials or promising therapeutic approaches targeting this pathway.

Overall, the manuscript titled "At the Crossroad of the cGAS-cGAMP-STING Pathway and DNA Damage Response: Implications for Cancer Progression and Treatment" by Korneenko et al. is a well-written and comprehensive review of the cGAS-cGAMP-STING pathway and its significance in cancer biology. The authors provide a thorough discussion of the pathway's dual roles in cancer and highlight the challenges and potential breakthroughs in developing therapeutic interventions. I recommend this manuscript for publication with the suggested revisions.
